# Boron-Doped Reduced Graphene Oxide with Tunable Bandgap and Enhanced Surface Plasmon Resonance

**DOI:** 10.3390/molecules25163646

**Published:** 2020-08-11

**Authors:** Muhammad Junaid, M. H. Md Khir, Gunawan Witjaksono, Nelson Tansu, Mohamed Shuaib Mohamed Saheed, Pradeep Kumar, Zaka Ullah, Asfand Yar, Fahad Usman

**Affiliations:** 1Department of Electrical and Electronic Engineering, Universiti Teknologi PETRONAS, Seri Iskandar 32610, Perak, Malaysia; Pradeep.hitesh@gmail.com; 2Department of Electronic Engineering, Balochistan University of Information Technology, Engineering, and Management Sciences, Quetta 87300, Balochistan, Pakistan; 3BRI Institute, Jl. Harsono RM No.2, Ragunan, Passsar Minggu, Jakarta 12550, Indonesia; Gunawan.witjaksono@gmail.com; 4Center for Photonics and Nanoelectronics, Department of Electrical and Computer Engineering, Lehigh University, 7 Asa Drive, Bethlehem, PA 18015, USA; Tansu@lehigh.Edu; 5Department of Mechanical Engineering, Universiti Teknologi PETRONAS, Seri Iskandar 32610, Perak, Malaysia; Shuaib.saheed@utp.edu.my; 6Department of Fundamental and Applied Sciences, Universiti Teknologi PETRONAS, Seri Iskandar 32610, Perak, Malaysia; Asfandyarhargan@gmail.com (A.Y.); Fahad_17004459@utp.edu.my (F.U.)

**Keywords:** graphene oxide, microwave, hydrothermal, boron-doped reduced graphene oxide, oxygen reduction reaction, optical bandgap, surface plasmons resonance

## Abstract

Graphene and its hybrids are being employed as potential materials in light-sensing devices due to their high optical and electronic properties. However, the absence of a bandgap in graphene limits the realization of devices with high performance. In this work, a boron-doped reduced graphene oxide (B-rGO) is proposed to overcome the above problems. Boron doping enhances the conductivity of graphene oxide and creates several defect sites during the reduction process, which can play a vital role in achieving high-sensing performance of light-sensing devices. Initially, the B-rGO is synthesized using a modified microwave-assisted hydrothermal method and later analyzed using standard FESEM, FTIR, XPS, Raman, and XRD techniques. The content of boron in doped rGO was found to be 6.51 at.%. The B-rGO showed a tunable optical bandgap from 2.91 to 3.05 eV in the visible spectrum with an electrical conductivity of 0.816 S/cm. The optical constants obtained from UV-Vis absorption spectra suggested an enhanced surface plasmon resonance (SPR) response for B-rGO in the theoretical study, which was further verified by experimental investigations. The B-rGO with tunable bandgap and enhanced SPR could open up the solution for future high-performance optoelectronic and sensing applications.

## 1. Introduction

Surface plasmon resonance (SPR) biosensors are optical sensors that have attracted greater attention due to their advantages such as high sensitivity, room temperature real-time measurements, and non-invasive measurements [1]. SPR is extremely sensitive to the alteration of the refractive index at the metal-dielectric interfac [2]. As such, sensitivity is exploited in the detection by any change at the interface due to the presence of an analyte. Gold is typically used as a resonant layer in SPR biosensors due to its chemical stability, where gold exhibits a higher shift of resonant angle with the change in refractive index [3]. However, the inertness of gold makes the detection or adsorption of the biomolecules very difficult at low concentrations, and sometimes even impossible [4,5,6]. Specifically, the single-crystalline gold surface does not measure the adsorption of small molecules (such as CO, H_2_, O_2_, NO, NH_3_) under high vacuumed (UHV) conditions or at elevated pressures and temperatures [5]. In numerous studies, the chemically and physically modified gold layer has been investigated for enhanced adsorption and detection mechanisms, such as stepped/kinked gold surface [7] gold nanorods, gold nanocrystal, CNTs [8] and graphene oxide coupled with gold nanoparticles [9].

Some novel materials such as graphene [10], reduced graphene oxide (rGO) [11], Au/Ag/Au/chitosan-graphene oxide [12], and MOS_2_ [13] have been explored to enhance the sensitivity and selectivity of SPR sensors. Among them, graphene has emerged as the most promising material owing to its unique properties, such as a high mobility and high surface-to-volume ratio that are beneficial for the efficient adsorption of biomolecules compared to gold. Similarly, the doped graphene oxide (GO) enriched with abundant defect sites on top of a gold layer is also presumed to exhibit an enhanced SPR response [14]. However, the modulation of the optical bandgap and Fermi energy of GO caused by chemical doping result in changes in the optical properties such as refractive index and dielectric constants. These variations could be monitored by SPR sensing techniques. 

Recent studies have shown significant improvements in graphene properties by doping it with different heteroatoms such as Florine (F), Oxygen (O) [15], Nitrogen (N) [16], and Boron (B) [17]. The boron-doped graphene oxide has been reported in various applications such as supercapacitor [18], Li-ion batteries [19], solar cells [20], and the enhanced photogenerated catalysis [21]. Moreover, several techniques such as hydrothermal method [22], arc-discharge process [23], and chemical vapor deposition (CVD) [24] have also been employed for the doping of graphene-based materials, where stable atomic substitution with carbon are highly desirable for the tunability of optical and electronic properties of graphene [25]. Among the reduction methods, GO is mostly reduced thermally at high temperatures [26]. However, the reduction at high temperatures produces high-density defects on the edge and basal planes, which deteriorates the electrical and the optical properties [27].

In this work, we have investigated the plausibility of microwave-assisted hydrothermally grown boron-doped reduced graphene oxide (B-rGO) as a promising material to achieve desired properties. The reduction of GO using the microwave-assisted technique is favorable owing to its easy process-ability, short time requirement, high yield, non-requirement of hazardous chemicals, and specialized equipment [27]. The chemical substitutional doping of boron atoms in the sp^2^ bonded hexagonal structure of the carbon network induces P-type characteristics with an enhanced hole concentration and charged carrier’s mobility [28]. The boron doping of graphene also promotes excellent electronic properties by the tunability of the energy bandgap and electrical conductivity [18]. We investigated the optical bandgap tuning capability of B-rGO and its application in surface plasmon resonance. B-rGO was initially synthesized using boric acid via the microwave-assisted hydrothermal technique. The structural and morphological studies of the synthesized B-rGO were then performed using the FESEM, FTIR, XPS, Raman, and XRD techniques. The bandgap study was conducted using UV-Vis spectra and Tauc plot estimations. The SPR responses were investigated experimentally using customize Kretschmann configuration and compared with the corresponding simulated results. The obtained results showed an improvement to the optical bandgap, electrical conductivity, and SPR sensing performance of the B-rGO compared to GO.

## 2. Results and Discussion

### 2.1. FESEM Analysis

Figure 1a,b shows the FESEM image of B-rGO consisting of a planar-like structure. The layered-shape surface of B-rGO contains several wrinkles and corrugations, possibly due to the presence of boron atoms and oxygen-related functional groups. The elemental compositions of B-rGO are presented in its dispersive energy X-ray (EDX) spectra, as shown in Figure 1c, indicating the strong signature of carbon (C) atoms. The signatures of boron (B) and oxygen (O) are also visible but not discretely recognized, possibly due to their near-atomic number with C and small concentration. The presence of a few unmarked peaks indicates the presence of platinum (Pt) [29]. The element mapping was also carried out to confirm the element distribution of B-rGO, as shown in Figure 1d–f for elements C, O, and B, respectively. The homogeneous dispersion of the three identified elements was found to conform in the selected area of 30 µm^2^.

### 2.2. FTIR Spectroscopy

The synthesis of GO and B-rGO was also tracked using FTIR spectra analysis, as shown in Figure 2a. On account of the GO analysis, a broad FTIR spectrum in range 3700–3000 cm^−1^ was observed that corresponds to the presence of a hydroxyl group (H_2_O and COOH possibly), O–H starching vibrations, and adoption of water molecules [30,31]. The corresponding peaks at 1226, 1418, and 1743 cm^−1^ are attributed to the epoxide (C–O–C), carboxyl COOH/H_2_O), and ketone (C=O) groups, respectively [32,33]. However, the (OH) bending vibration at 1620 cm^−1^ contributed due to the absorption of water molecules, which is also attributed to the (C=C) aromatic vibration [34]. 

Furthermore, the peak at 1043 cm^−1^ was assigned to the stretch vibration of C–O groups [35]. Subsequently, after microwave-assisted hydrothermal reaction of GO and boric acid, the peaks correlated with the oxygen-containing group were found to be substantially reduced, and even some disappeared. Moreover, new additional peaks were observed in the FTIR spectrum range from 1,000 to 1750 cm^−1^. Therefore, the vibrational band 1735 cm^−1^ was assigned to the C=O functional group, which was slightly shifted to the 1695 cm^−1^ with lower intensity as compared with the GO spectrum. In the FTIR spectrum of B-rGO, the band around 1040 cm^−1^ ascribed to the asymmetric B–O–B stretch B-O bond among one tetrahedral and one trigonal B atom [36]. Usually, the B–C functional group was observed from 1050 to 1200 cm^−1^, where the peak at 1180 cm^−1^ corresponded to the stretching vibration of the B–C bond. Particularly, the effective downshift in the C network was observed due to the B substitution, which was ascribed to the decreased force constant in the stretching vibration of the B–C bond in comparison with C–C, comparatively [36]. However, the existence of higher C content might have induced the enhanced frequency of B–C stretching [37]. 

### 2.3. Raman Spectroscopy

Figure 2b shows the Raman spectra of GO and B-rGO samples at ambient temperatures with 514 nm laser excitations, where the Raman measurement was carried out from the 0 to the 4000 cm^−1^ spectrum region. From the Raman spectra of GO, the intensity peaks found at 1349 cm^−1^ and 1601.17 cm^−1^ were ascribed to the D- and G- bands, respectively. The D-band at ~1349 cm^−1^ observed in the GO spectrum, was correlated to the defects and disorders in the graphitic lattice, due to the addition of oxygen-related functional groups. In addition, the second-order D, also known as 2D or G’, appeared at 2688 cm^−1^, attributing to the process of the two second-order phonon vibrations [19]. After the boron doping, a slight redshift in the G-band was observed in B-rGO samples in comparison to GO [38,39]. However, the increment in the D-band intensities in comparison with the G-band intensities of the B-rGO spectra indicated that the extended deformation exhibited the boron doping. Moreover, the intensity ratio (*I_D_/I_G_*) defined the quality of the material and the relative concentration of sp^3^ defects in the sp^2^-hybridized graphene structure [21]. An increment in (*I_D_/I_G_*) ratio was also observed in the B-doped configurations, i.e., GO (0.94), B-rGO1 (1.094), B-rGO2 (1.12), and B-rGO3 (1.137). The increase in the *I_D_/I_G_* ratio after microwave-assisted hydrothermal treatment indicated the reduction of GO with the existence of boron atoms. The shift of the disorder and graphitic (G+D) peaks at 2934, 2933, and 2936 cm^−1^, respectively, indicated the existence of few-layer graphene [40]. The shift of the 2D band in the B-rGO spectra also specified the weak (B-C) carbon bonding of 2.57 eV in comparison with the 3.71 eV of (C–C) bonding [41].

Furthermore, the electrical conductivity measurement was carried out to integrate the Raman information. The electrical conductivity of GO and B-rGO samples was measured at room temperature using Hall-effect measurement (HMS-3000 Series) (see Appendix A). It was observed that the electrical conductivity of GO increased with the reduction of the oxygen-related functional groups and an increase in sp^2^ domains. The electrical conductivity of GO was found to be further improved with boron doping, and this might have been dependent on the doping concertation of the incorporated atoms [35]. The increment in the p-type electrical conductivity corresponded to the increase in boron contents in the B-rGO samples. These results were found to be in agreement with previous studies [27,42].

### 2.4. XPS Analysis

The XPS survey scan for GO and B-rGO samples is shown in Figure 3a, which indicates the C1s, O1s, and B1s peaks at 285, 533, and 190 cm^−1^, respectively. Moreover, reduced O1s peak in the B-rGO survey indicated the reduction of oxygen-related groups in the GO-modified structure. Figure 3c shows the Gaussian deconvolution of the O1s spectrum that could be split into four peaks 531, 532, 533, and 534 eV, confirming to the O=C–OH, C–OH, and C–O, functional groups, respectively [43]. Moreover, the atomic percentage of each element was also determined from the XPS spectrum analysis of GO and each of the three B-rGO samples, as summarized in Table 1. In these studies, the atomic percentage of boron contents in the B-rGO samples was found to be 4.80, 5.53, and 6.51 at.%, respectively, showing an agreement along with the previously reported work [19,44,45]. The maximum boron doping concentration, up to 6.51 at.% was confirmed through XPS analysis, where B-rGO samples with lower doping could also be synthesized by varying the concentration of boric acid and GO in the synthesis process. More interestingly, a decrement in the atomic percentage of oxygen was observed, where the atomic percentage of oxygen for the GO, B-rGO1, B-rGO3, and B-rGO3 samples was determined to be 34.15, 11.46, 10.27, and 9.58 at. %, respectively. Table 2 presents each functional group determined from the deconvolution of the C1s, O1s, and B1s XPS peaks, respectively. 

The deconvolution of the C1s spectra in the B-rGO indicated several peaks comprising of C=C (285 eV), C–OH (286 eV), and C=O (288 eV), related to the various chemical bonding carbon in the reduced graphene oxide structure. Especially, an intense peak in the C1s spectra at 284.75 eV corresponded to the C–C sp^2^ bonded graphite carbon structure. It was also implied that the majority of carbon atoms were in honeycomb structure [19]. Figure 3d illustrates the XPS spectrum B1s of B-rGO. The Gaussian deconvolution of the entire B1s spectrum showed that the entire spectra could be deconvoluted in five peaks sequentially. The B1s peak at ~191 eV was found to be higher than the pure boron peak of 188 eV [46], indicating the existence of boron in the oxidation states. The deconvoluted peaks of the B1s also indicated the B_4_C (187 eV), BC_2_O_2_ (188 eV), BC_2_O (190 eV), and BCO_2_ (193 eV) peaks, confirming the successful doping of the GO carbon network with boron atoms [35]. The B-sub-C (188 eV) deconvoluted peak referred to the substitution doping of B atoms with C atoms in the graphite structure [47]. The XPS study of B-rGO revealed that GO was not fully reduced, and the synthesized B-rGO samples still contained some oxygen-related functional groups.

### 2.5. XRD Analysis

The crystalline structure of GO and boron B-rGO has been investigated by the XRD diffraction pattern. Figure 4 shows the XRD pattern of the synthesized GO, doped B-rGO1, B-rGO2, and B-rGO3 powder samples. The (001) strong and diffraction peak for GO was observed at 11° = 2θ [48]. After boron doping, the (001) peaks had disappeared, and the XRD diffraction peaks (002) for B-rGO1, B-rGO2, and B-rGO3 appeared at 25.93°, 25.95°, and 26° = 2θ, respectively, showing an agreement with previously reported data [49]. Also, the lower XRD peaks (001) were also observed for all B-rGO samples at 43°.

Moreover, the interlayer distance between the wave field and the crystalline plane was determined by using Bragg’s Equation (1) [50],
(1)$$\lambda =\frac{2d\sin\left(\theta \right)}{n}$$where, *θ*, *λ, n*, and *d* are the scattering angle, the wavelength of the incident x-ray (1.54 Å), the order of reflection, and the interlayer distance, respectively. The interlayer distance of GO ~8.05 Å showed an agreement with reported work [51,52]. Higher layer distance in the synthesized GO was observed due to the repulsive forces induced by the presence of oxygen-related functional groups and the water molecules. Moreover, the interlayer distance of 4.21 Å, 4.18 Å, and 4.15 Å were determined for the B-rGO1, B-rGO3, and B-rGO3, respectively. The decrement in the interlayer distance, as observed in the XRD spectra of B-rGO, was attributed to the increment in boron concentration and also resulted from the decrement in a considerable amount of oxygen-related functional groups [35].

### 2.6. Optical Bandgap Studies

The optical bandgap of B-rGO samples was extracted using ultraviolet−visible (UV-Vis) absorption spectral analysis and the Tauc plots, as shown in Figure 5. UV-Vis spectra of all B-rGO samples (Figure 5a) display the absorption peaks from 375, 382, and 386 nm due to π to π* transitions, which corresponds to the direct optical bandgap ~2.91 to 3.05 eV [53]. The lower light absorption was observed from 450 to 800 nm, attributing to the maximization of sp^2^ domains in the B-rGO. Moreover, the extra shoulder in the UV-Vis spectra observed from 375 nm to 386 nm for B-rGO samples correlated to the n to π* transitions C=O bonds [54]. This corresponded to the electrons transition from boron states to the π* states, indicating the heterogeneous distribution of boron atoms in the carbon network of reduced graphene oxide. It was also interesting to note that the UV-VIS absorption spectra of the B-rGO samples showed strong absorption of light in the region of the UV spectrum towards almost visible.

The optical bandgap sketched for GO, B-rGO1, B-rGO2, and B-rGO3 is shown in Figure 5b to 5e, where the value of the direct energy bandgap was inferred by extrapolating the linear part of the UV-Vis absorption curve to zero x-axes. From the Tauc plot calculation based on Equation (6), the direct bandgap values for the GO, B-rGO1, B-rGO2, B-rGO3 samples were determined to be ~3.41, 3.05, 2.94, and 2.91 eV, respectively. It was also determined that the B-rGO3 samples with a maximum boron doping concentration of 6.51 at. % depicted the lowest optical bandgap value (2.91 eV), comparatively. Moreover, the direct optical bandgap was further investigated with the increment of boron doping concentration, where the bandgap was found to be reduced, attributing to the reduction of the oxygen-related functional groups in the rGO carbon network. The observed bandgap reduction could also be ascribed to the compensation of energy states associated with the incorporation of boron dopant atoms, that ultimately shift the direction of the conduction band edges [55]. 

### 2.7. Surface Plasmons Resonance (SPR) Study of B-rGO

The optical constant for the boron-doped reduced graphene thin film samples (B-rGO1, B-rGO2, and B-rGO3), particularly the real part of the dielectric constant and complex refractive index was determined by applying Equations (2)–(11) at 633 nm, as shown in Table 3. Where the 633 nm laser source was conventionally used for gold-based SPR sensing applications [56]. Besides, the values of the optical constants of the gold and the glass prism (SF11) are also included in Table 3 for the SPR studies. Prior to the SPR study of B-rGO, the thickness of the Au layer was optimized for the minimum reflectance *R_m_*, as presented in Figure 6a. The optimization of *R_m_* was performed to ensure the maximum surface plasmon excitation. The optimal SPR response with the deepest curve was determined from the gold thin film with 50 nm thickness, as shown in Figure 6a, while the same thickness value was considered for the later studies.

The constant optical values of the B-rGO samples were decreased in comparison to that of GO and rGO, as shown in Table 3, attributing to the possible reduction of oxygen-related functional groups in the B-rGO after doping. On the other hand, the depth and width of the SPR reflectivity curve were affected by the change in the values of the optical constants and the thickness (d) of the light-absorbing surface (gold surface), as shown in Figure 6b [59]. It could be observed that the SPR curves for B-rGO1, B-rGO2, and B-rGO3 were overlapped due to the marginal difference in their constant optical values. To assess the SPR sensing performance of GO and B-rGO samples, the parameter FWHM, and the value of the penetration depth through the dielectric medium *δ_d_* were investigated. Particularly, the high-performance SPR sensing layer was expected to possess low FWHM and high *δ_d_* values [2]. Figure 6c shows the variation of the FWHM for the gold, GO, thermally reduced GO, B-rGO1, B-rGO2, and B-rGO3-based SPR sensors, attained using Equation (18). GO showed the lowest FWHM response as compared with B-rGO samples (see Table 3). In Figure 6d, the *δ_d_* relative values calculated from Equation (19) for the gold, GO, thermally reduced GO, B-rGO1, B-rGO2, and B-rGO3 samples were found to be 359.64 nm, 103.82 nm, 100.57 nm, 128.85 nm, 128.16 nm, and 127.63 nm, respectively. 

The measured thickness of the Au and B-rGO layers was found to be 50 ± 1.5 nm and 5± 1 nm, respectively. In Figure 6e, the experimental and simulated SPR responses of the B-rGO1, B-rGO2, and B-rGO3 were compared. The position of the SPR angle for the measured and simulated B-rGO1, B-rGO2, and B-rGO3-based SPR sensors were found to be almost the same. However, the SPR curves for the simulated sensors were deeper as compared to the measured values. This could be attributed to the slight variation in the thickness of the Au/B-rGO layer and the refractive index values of the SF11 Prism and index matching liquid. Interestingly, the B-rGO demonstrated the highest values of δd, compared to the GO, and the thermally reduced rGO. This shows that B-rGO-based sensors could produce an enhanced SPR response.

## 3. Materials and Methods

### 3.1. Materials

Graphite flakes (~50 μm), ethanol (95.0%), hydrochloric acid (HCl) (37%), sulfuric acid (H_2_SO_4_) (97%), phosphoric acid (H_3_PO_4_) (85%), potassium permanganate (KMnO_4_) (99.5%), hydrogen peroxide (H_2_O_2_) (30%), and other analytical grade reagents purchased from R&M, Selangor, Malaysia. Boric acid (ACS reagent, 99.5%, Darmstadt, Germany) was purchased from Sigma-Aldrich. Graphene oxide (GO) was synthesized by improved Hummer’s method [60], and the boron-doped-reduced graphene oxide was produced by the facile microwave-assisted hydrothermal method using boric acid.

### 3.2. Synthesis of Graphene Oxide

Initially, 360 mL H_2_SO_4_ and 10 mL H_3_PO_4_ were mixed and stirred for 2 h in a 1000 mL round bottom flask. Three grams of graphite flakes and 18 g potassium permanganate were then added to the mixed solution (H_2_SO_4_ + 10 mL H_3_PO_4_), which formed a dark green solution. The process was carried out in an ice bath to prevent overheating and explosion. The existing solution was stirred for 12 h to oxidize and exfoliate the graphite into a few layers of GO structure. After the completion of the reaction, H_2_O_2_ was added to stop the further reactions. At that point, the solution color was changed into a dark yellow-brown. Furthermore, 1 L of deionized (DI) water was used to dilute the solution, and then its pH level was adjusted ~1. The obtained mixture was then centrifuged at 5000 rpm to collect the graphite oxide and decant away the acid. The obtained oxidized graphite flake was further washed with the 10% HCl to remove the impurities like potassium and manganese ions, which are typically trapped in the solid. Afterward, the remaining mixture solution was washed numerous times with DI water while waiting for the pH level to reach ~1. GO flakes were then collected by a centrifugation process at 10,000 rpm. The GO flakes were dried in a Freeze Dryer and ground into a fine powder. At this stage, the powder was exfoliated into a few and single layers, so it was referred to as graphene oxide (GO) powder. 

### 3.3. Synthesis of B-rGO

The GO solution (70 mg/100 mL) was prepared in DI water followed by continuous stirring for 2 h and sonication for 0.5 h (Scheme 1).One to 3 g of boric acid solution was also prepared in 10 mL ethanol by magnetic stirring at 60 °C. After uniform dispersion of boric acid in ethanol, both obtained solutions were mixed and stirred for 8 h at 80 °C. Subsequently, the obtained mixture solution was poured into 200 mL Teflon tube autoclave and heat-treated in the oven at 180 °C for 12 h. The obtained B-rGO flakes were further exposed to microwave energy radiation using the 800 W microwave oven for the 40 s. The intensity of the microwave intensity was adjusted to an optimized level to enhance the further reduction and absorption of unreacted boron contents. The resulting B-rGO powder was washed several times with DI water to remove untreated boric acid contents. Finally, the dry B-rGO powder was obtained by the dry freezing process in a vacuum oven at 60 °C for 36 h. The three samples of boron-doped reduced graphene (B-rGO1, B-rGO2, and B-rGO3) were synthesized with three different concentrations of boric acid, 1, 2, and 3 g, respectively. 

### 3.4. Material Characterizations

Morphology of the B-rGO, with energy dispersive X-ray (EDX) component mapping, was examined by field emission scanning electron microscopy (FESEM, Zeiss Supra 55 VP, Cambridge, UK). The functionalized surface of the GO and B-rGO thin film was studied by Fourier transform infrared (FTIR) spectroscopy (Bruker Instruments, model Aquinox 55, Ettlingen, Germany). Raman spectroscopy (Bruker Instruments, Horiba Jobin Yvon HR800, Ettlingen, Germany) was also performed to reconfirm the arrangement of the composites. The imperfections and defects in the GO and B-rGO samples were also analyzed using Raman spectroscopy. In Raman spectra of GO and B-rGO, various peaks with different intensities appeared, where each peak to the material, followed the imprints and grouping of deformities in materials.

The optical characterization of spin-coated GO and B-rGO thin film was performed by UV-Vis Spectrophotometer (Cary 100, Agilent Technologies, Santa Clara, CA, USA) at room temperature. The thickness estimation was obtained from directly utilizing the surface harshness analyzer (SV-Mitutoyo-3000, Aurora, IL, USA). The XRD estimation was also conducted for additional affirmation of crystallinity and synthesis of GO and B-rGO powder samples by utilizing XRD (X’Pert3 Powder, Westborough, MA, USA). The XRD analysis was performed over a 5–90° range with step size 0.01/s. Likewise, the existence of various functional groups was also confirmed through the support of X-ray photoelectron spectroscopy analysis (XPS; Thermo logical, K-alpha, Waltham, MA USA). The XPS survey spectra were recorded from 0 to 1200 eV, under pressure conditions of 10^−8^ mbar through monochromatic Al Ka radiation (1486.6 eV). The XPS spectra were further analyzed with the help of the Origin (version 9.0, Waltham, MA, USA) software. The deconvolution of multiple peaks was performed on XPS raw data. The existence of individual functional groups in XPS spectra was also analyzed from multiple peaks by the implication of Gaussian curve fittings, where the estimation of the baseline was performed by utilizing the adjacent-averaging technique. The electrical conductivity was measured at room temperature using a Hall Effect measurement (HMS-3000 series, Phoenix, AZ, USA).

### 3.5. Optical Bandgap and Optical Constant Measurement

The reduced optical bandgap energy of B-rGO indicates the prospective utilization of the material in optoelectronic device applications. The optical bandgap and optical constants of the B-rGO samples were obtained from the UV-Vis spectrum of a thin film. The GO and B-rGO dispersion solutions in N-Methyl-2-pyrrolidone (NMP) with a concentration of 0.25 mg/mL were used for the thin film deposition. The process was complemented by sonication or 0.5 h followed by the strong magnetic stirring for 2 h. Before spin coating, the obtained dispersed solution was centrifuged at 6,000 rpm to segregate the undissolved B-rGO particles. Later, B-rGO based thin film was prepared by using a spin coater (POLOSTM) at 3,000 rpm. From a theoretical perspective, the relationship among the parametric values obtained from UV-Vis spectroscopy, namely; absorption (A), transmission (T), and reflectance (R) could be defined by Equation (2) [61].
(2)R+T+A=1(A=1−R−T)

The Bear–Lambert law (Equation (3)) establishes the relationship between light attenuation through the specific material and its properties, where the relation between transmittance and absorbance of light could be defined [62]. The transmittance and absorption coefficient could be determined by using Equations (4) and (5), accordingly [63].
(3)$$I\;\left(d\right)=I_{o}e^{-ad}$$
(4)$$T={\left(1-R\right)}^{2}e^{-ad}$$
(5)$$a=\frac{2.303A}{d}$$
where *I(d)* is the intensity at a depth of thickness *d*, *I_o_* is the intensity at zero thickness, and *a* is the absorption coefficient. The optical bandgap structure and the type of electronic transitions could be obtained from the absorption coefficient [64]. The optical bandgap (*E_g_*) could be determined by using Equation (6).
(6)$$E_{g}=hv-{\left(\frac{ahv}{\beta }\right)}^{\frac{i}{j}}$$
where *hv* is the photon energy, *E_g_* is the energy bandgap; *a* is the absorption coefficient, and *β* is a constant for disorder parameter which is independent of photon energy. The value of parameter *j* denotes the nature of transition, where the parametric value of *J*, 1/2 used for direct allowed transitions, and 3/2 used for the direct forbidden transitions. However, the *j* parametric value of 2 is described for indirect allowed transitions, and 3 defines the indirect forbidden bandgap transitions [65]. The boron-doped graphene possesses direct electronic transitions [66]. Therefore, in this study, the optical bandgap was determined from the Tauc plot estimation, i.e., the optical bandgap (*hv*) verses (*ahv*)^2^. Furthermore, the imaginary part of the refractive, *K* (extinction coefficient) correlated with an absorption coefficient by the Equation (7).
(7)$$k=\frac{aʎ}{4\pi }$$
where *λ* is the wavelength of the light wave. Based on the UV-Vis reflectivity measurement, the real part of the refractive index (*n*) could also be obtained from the Fresnel Equation (8) based on the reflectivity measurement using Equation (9).
(8)$$R=\frac{{\left(n-1\right)}^{2}+k^{2}}{{\left(n+1\right)}^{2}+k^{2}}$$
(9)$$n=\left[\frac{1+R}{1-R}\right]+\sqrt{\frac{4R}{\left(1-R\right)}}-k^{2}$$The value of the complex dielectric of B-rGO could also be evaluated from the value of *K*, and *n*.
(10)$$\varepsilon_{r}=n^{2}+k^{2}$$
(11)$$\varepsilon_{i}=2nk$$
where *ε_r_* and *ε_i_* are the real and imaginary parts of the B-rGO dielectric constant, respectively.

### 3.6. Surface Plasmons Resonance (SPR) Study

The surface plasmons resonance of the prepared samples was studied using our customized system, which was entirely based on the Kretschmann configuration, as shown in Figure 7a. The polarizer is an optical filter that is utilized to P-polarize the incident light beam in the direction parallel to the plane of incidence. However, in this case, the direction was within the horizontal plane. The modified Kretschmann configuration was used to excite the surface plasmon in which metallic film was in direction with a prism base. The chopper (SR 540) and lock-in amplifier (SR 530) was incorporated within the proposed system to optimize the measuring result in terms of signal to noise ratio (SNR). Besides, in this configuration, the chopper was used for the intensity modulation of the incident light, and the lock-in amplifier was used to measure the corresponding data to the chopping frequency. To evaluate the change over the momentum in incident light regarding the dielectric-metal interface, the prism (*n* = 1.77861) was fixed on the rotation table and circled with a step size of 0.1° using a stepper motor (Newport MM 3000, Scotia, NY, USA). In this system, the He–Ne (633 nm, 5 mW) laser beam was utilized with a photodiode detector. 

The existence of resonating free electrons with the collective oscillation on the plasma surface (i.e., metal surface) may induce charge density transverse waves propagation along the plasma surface. The oscillation of the transverse wave normal to the plasma surface is called surface plasmons. The surface plasmons resonance gets excited when the electromagnetic waves are coupled through the particular medium on the metallic surface. The light source must be P-polarized to the plasma surface, where the SPs only have the electric field component normal to the surface [67]. Consequently, the SPs get excited when the momentum between the surface plasmons (SPs) and the incident light is matched. The maximum energy transfer between SPs and photons occurs at the SPR angle, which mainly depends on the dielectric property of adjacent material on the gold thin film surface, and it is stated by *K_sp_* (surface plasmons vector) as shown in Equation (12).
(12)$$K_{sp}=\frac{\omega }{c}\;\sqrt{\frac{\varepsilon_{1}\varepsilon_{2}}{\varepsilon_{1}+\varepsilon_{2}}}$$
where *c* is the velocity of light, *ω* is the frequency and $\varepsilon_{m},\varepsilon_{d}$ are the dielectric constant for the metal and dielectric material, respectively. The electric field is perpendicular to the Au-coated thin film. The relationship between the dielectric constant and refractive index could be defined by Equation (13).
(13)$$\varepsilon =n^{2}$$
Equation (12) can be rewritten as,
(14)$$K_{sp}=\frac{2\pi }{\lambda }\;\sqrt{\frac{n_{2}{}^{2}n_{g}{}^{2}}{n_{2}{}^{2}+n_{g}{}^{2}}}$$

Furthermore, the total internal reflection (TIR) occurs when the light waves travel from a material with a higher refractive index to another material with a lower refractive index. For TIR, the incident angle (θ) must be greater than the critical angle (*θ_c_*), therefore *θ_c_* = sin^−1^(*n*_2_/*n*_1_). Likewise, the evanescent waves induced within the medium with a lower refractive index, the induced waves decay exponentially to the interface distance between both mediums. Therefore, the evanescent wave vector (*Kevan.* ‖) can be determined by Equation (15).
(15)$$k_{evan.\|}=\frac{2\pi }{\lambda }n_{1}\sin\left(\theta \right)$$
(16)$$k_{evan.\|}=K_{SP}$$
where *θ*, *n*_1_, and *λ* are the incident angle, the refractive index of gold, and the wavelength of the incident light, respectively. When the evanescent wave excites the surface plasmons, the intensity of reflected light decreases sharply. The decrease in excited plasmons induces the energy conversion to photons and phonons. Therefore, the *K_SP_* should be equal to *K_evan.__‖_*. According to the requirement of the phase-matching condition for SPR excitations [68]; *θ_SPR_* angle as a function of related refractive index could be obtained by solving Equations (14)–(16) together (Equation (17)).
(17)$$\theta_{spr}=\sin^{-1}\left(\frac{2\pi }{n_{1}}\sqrt{\frac{n_{2}n_{g}}{n_{2}+n_{g}}}\right)$$

Specifically, the SPR angle of the incident P-polarized light mainly depends on *n_1_*, *n*_2_, and *n_g_*. The modification in the refractive index of the second medium near the dielectric and metal interface and the change in resonance angle also take place, accordingly. Hence, the shift in the *θ_SPR_* angle could be used to assess the variation induced by the change in the refractive index of plasmonic material. Based on the above equations, the SPR reflectivity curve was also simulated using the in-house developed MATLAB simulation program. The program allows the flexible designing of the Kretschmann SPR configuration based on the values of the optical constants and thicknesses of the material of interest, as shown in Figure 7b. The improved SPR response of the B-rGO samples was investigated by comparing the detection accuracy value or the signal to noise ratio (SNR) of the bare gold, GO, and B-doped GO. The sensitivity and the SNR were calculated using Equation (18) [2], where SNR is explained in terms of FWHM.
(18)$$SNR=\frac{\Delta \theta_{spr}}{FWHM}$$
where, Δ*θ_SPR_* and FWHM are the SPR angle shift and the full-width-at-half maximum, respectively. Besides, the penetration depth through the dielectric medium (*δ_d_*) were also considered as an essential parameter, which describes the length over which the surface plasmon was sensitive to the changes in the refractive index in the dielectric medium. It could be calculated using Equation (19).
(19)$$\delta_{d}=\frac{\lambda_{0}}{2\pi }\left[\frac{\varepsilon_{m}+\varepsilon_{d}}{\left(\varepsilon_{d}\right)}\right]{}^{1/2}$$
where, *λ_0_* is the free space wavelength, *ε_m_*, and *ε_d_* are the real part dielectric constants of gold and the adjacent material to the gold, correspondingly [2].

### 3.7. Fabrication of Au/B-rGO Sensor Chips

The gold-coated glass (AU. 0500. ALSI, City, Fitchburg, WI, USA) with a uniform thickness of 50 nm Au layer was purchased from Platypus Technology, USA. The thickness of the Au layer was further confirmed by the surface roughness tester (SV 3000, Mitutoyo, Aurora, IL, USA). The B-rGO dispersion solution (0.25 mg/mL) in NMP was attained after the 1 h sonication process, using the probe sonicator. After that, the unexploited flakes of B-rGO were removed by the centrifuge process at 6000 rpm for 10 minutes. The Au-coated glass was washed with DI water before the deposition of the B-rGO layer. The uniform B-rGO thin film was attained by using a spin-coating technique at the rotation speed of 4,500 rpm followed by its thickness measurement using the surface roughness tester. Later, the sensing layer coated on top of the gold film was attached to the SF11 prism using the Norland index matching liquid, Newport to prepare the sensor chip.

## 4. Conclusions

In this work, we studied the enhanced SPR response with a tunable optical bandgap in the visible region of B-rGO. The B-rGO was synthesized using a customized microwave-assisted hydrothermal approach. The successful synthesis of B-rGO was confirmed by standard FESEM, FTIR, RAMAN, XRD, and XPS-based analyses. The optical bandgap and optical constants for the B-rGO were extracted from the UV-Vis absorption spectrum. The direct bandgap for the three different B-rGO samples was found to be ~2.91, 2.94, and 3.05 eV, whereas the optical bandgap for GO was found to be 3.41 eV. Interestingly, the enhanced P-type electrical conductivity with reduced optical bandgap was also observed. The surface plasmon resonance-based sensing response of the samples was investigated theoretically as well as experimentally using the customized Kretschmann configuration and found to be enhanced for B-rGO. The acquired results validate the tunability of the optical bandgap and the enhanced surface plasmons resonance in B-rGO. The comparative depth penetration response obtained for B-rGO samples, predicts its utilization in future highly sensitive SPR-based sensors. The tunable optical and electronic properties of B-rGO could be further optimized for optoelectronic, sensor, and light-harvesting applications.

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
