# Peer review of "Boron-Doped Reduced Graphene Oxide with Tunable Bandgap and Enhanced Surface Plasmon Resonance"

_molecules, 2020, doi:10.3390/molecules25163646_

Round 1

Reviewer 1 Report

This manuscript reported the fabrication of boron doped reduced graphene oxide, and its tunable bandgap and enhanced surface plasmon resonance were characterized by a variety of methods. The increased conductivity and improved performance of surface plasmons resonance of B-rGO has great potential in optoelectronic and sensor applications. It is a thorough study, well-written and easy to understand.  Only minor changes are recommended.

  1. The text from line 324 to 339 are overlapped with Table 1(a) and Table 1(b).
  2. What are the differences between B-rGO1 to BrGO-3? I could not find any information in the experimental section.
  3. I think it is more obvious if authors can compare the atomic percentage difference of oxygen for GO and B-rGO for XPS result.

Author Response

Dear Reviewer,
Thank you very much for the valuable comments provided. We have addressed all the comments as shown in the revised manuscript (green line). We have restructured the paragraphs, added more text, references, redrawn and added figures and also expanded on the discussion.

On behalf of all co-authors,

Muhammad Junaid

Reviewer 2 Report

Title: Boron Doped Reduced Graphene Oxide with Tunable Bandgap and Enhanced Surface Plasmon Resonance

Manuscript ID: molecules-847574

Authors: Muhammad Junaid, M. H. Khir, Gunawan Witjaksono , Nelson Tansu, M.S.M. Saheed , Pradeep Kumar Kumar , Zaka Ullah, Asfand Yar , Fahad Usman

Manuscript Summary: A boron-doped reduced graphene was prepared and characterized by SEM, FTIR, XPS, Raman, UV-Vis, Conductivity Measurements and XRD. From the parameter obtained from the optical studies it is proposed that this material enhances the sensitivity in Surface plasmon resonance. This part seems to be evaluated only in theory.

Reviewers Comment:

  • The idea presented in this manuscript is novel and of significance to the development of SPR sensors surfaces. Nevertheless, the data presented regarding SPR sensing are not convincing at all.
  • The manuscript is well structured. Data from control experiments are missing. This leads to the fact that the interpretation of the data is somehow speculative.
  • Scholary presentation:
    1. Language needs to be improved
    2. The formatting of the manuscript is on some pages corrupt.
    3. Figures and tables should be presented right after the paragraph where they are discussed.
  • All data given should be treated statistically. None of the values given in the manuscript shows any error or standard deviation. This cannot be accepted in a scientific publication. At least all data should present the average of three independent measurements.
  • Not clear why electrical conductivity of the material was studied. It seems that it is only to be used to prove the reduction of GO to rGO. This was already seen from the Raman data and these results can be shifted to the supporting information.
  • What was the solvent to be used for the UV-Vis spectra shown in Figure 6? It seems that the authors used two different solvents for GO and B-rGO? Does this change in solvent have any impact on the peak position?
  • It remains unclear if the data for SPR have been achieved experimentally or only by calculation. If it was only done by calculation experimental verification need to be presented. If this data are obtained from SPR measurements a description in the experimental part is missing.
  • The statement “Despite the highest δd value for the gold-based SPR sensing layer, the poor adsorption capability of gold makes the B-rGO more feasible sensing material [4,5]. As such, the B-rGO based SPR sensor is expected to produce superior SPR response.” is overoptimistic. There is no information on the thickness of B-rGO on the gold? How is it possible to obtain a stable layer of homogeneous thickness? Gold has a very high adsorption capability for many biomolecules due to strong Au-S binding. That’s the reason why there are so many papers on SPR biosensors published. Gold can easily modified by thiol linkers to achieve sensor of high sensitivity.

Overall this manuscript is not convincing. Many details are given for the characterization methods, which is really good, but almost no data are presented for the SPR part from which the main conclusions have been drawn and which is topic of this work. Therefore I recommend to reconsider this manuscript after a major revision.

Author Response

(The authors gave the same response as above.)

Reviewer 3 Report

The authors in this article describe the preparation and the characterization of boron doped reduced graphene oxide and study especially optical bandgap and surface plasmons resonance. The manuscript is well written and documented although there are some mistakes in several sentences (like strikethrough words, double numbers, H2O etc). I suggest the publication of this article after some minor revisions.

  1. In the introduction the authors should include further applications related to Boron doped graphene from the literature.
  2. The authors should check again the expression of the equations of the paragraph 2.5 and 2.6.
  3. FTIR and Raman spectra of B-rGO should be better compared with rGO instead of GO.
  4. The same for XRD
  5. It would be interesting to have a B-rGO with higher Boron percentage like 10 % or a much lower like 1%, since the three products have rather similar Boron content. The authors should explain if the preparation method has the ability to produce highly doped rGO.

Author Response

Dear Reviewer,
Thank you very much for the valuable comments provided. We have addressed all the comments as shown in the revised manuscript (Red line).We have restructured the paragraphs, added more text, references, redrawn and added figures and also expanded on the discussion.

On behalf of all co-authors,
Muhammad Junaid

Round 2

Reviewer 2 Report

The authors have carefully revised the manuscript in accordance to the reviewer's suggestion. The quality of the manuscript was significantly enhanced. With this I recommend this work to be published in MDPI journal Molecules.